# Hybrid mass spectrometry approaches in glycoprotein analysis and their usage in scoring biosimilarity

Yang Yang[1,2,*], Fan Liu[1,2,*], Vojtech Franc[1,2,*], Liem Andhyk Halim[3], Huub Schellekens[3] & Albert J.R. Heck[1,2]

Many biopharmaceutical products exhibit extensive structural micro-heterogeneity due to an array of co-occurring post-translational modifications. These modifications often effect the functionality of the product and therefore need to be characterized in detail. Here, we present an integrative approach, combining two advanced mass spectrometry-based methods, high-resolution native mass spectrometry and middle-down proteomics, to analyse this micro-heterogeneity. Taking human erythropoietin and the human plasma properdin as model systems, we demonstrate that this strategy bridges the gap between peptide- and protein-based mass spectrometry platforms, providing the most complete profiling of glycoproteins. Integration of the two methods enabled the discovery of three undescribed *C*-glycosylation sites on properdin, and revealed in addition unexpected heterogeneity in occupancies of *C*-mannosylation. Furthermore, using various sources of erythropoietin we define and demonstrate the usage of a biosimilarity score to quantitatively assess structural similarity, which would also be beneficial for profiling other therapeutic proteins and even plasma protein biomarkers.

[1] Biomolecular Mass Spectrometry and Proteomics, Bijvoet Center for Biomolecular Research and Utrecht Institute for Pharmaceutical Sciences, University of Utrecht, Padualaan 8, 3584 CH Utrecht, The Netherlands. [2] Netherlands Proteomics Center, Padualaan 8, 3584 CH Utrecht, The Netherlands. [3] Department of Pharmaceutics, Utrecht Institute for Pharmaceutical Sciences (UIPS), Utrecht University, Universiteitsweg 99, 3584 CG Utrecht, The Netherlands. * These authors contributed equally to this work. Correspondence and requests for materials should be addressed to A.J.R.H. (email: a.j.r.heck@uu.nl).

Most human proteins are decorated by a plethora of post-translational modifications (PTMs) regulating their structure, function and interactions. Protein PTMs, such as phosphorylation, acetylation, ubiquitination, and glycosylation, have all been shown to regulate a wide range of biological processes. Protein glycosylation represents likely the most heterogeneous PTM owing to the plethora of co-occurring carbohydrate structures. Protein glycosylation regulates in particular signalling, cell–cell communication and regulation of host–pathogen interactions[1–3], providing valuable targets for inhibition of infection[4].

Over the last decade bottom-up mass spectrometry (MS) based proteomics has emerged as a powerful technique for protein PTM identifications, whereby proteins are first enzymatically digested and thereafter PTMs are identified at the peptide level[5–7]. In bottom-up MS approaches, tandem MS (MS2) fragmentation can be effectively performed on peptides, enabling the characterization of PTM site-localizations and the structure of modifications[8]. A downside of bottom-up approaches, however, is that the peptides analysed contain mostly single PTMs hampering the investigation of the whole-protein picture. Especially in recent years, with the novel strategies of glycopeptide enrichment and hybrid MS fragmentation, bottom-up MS is widely applied to glycopeptide characterization[9–13]. Since glycopeptides are on average much larger than their non-modified counterparts (arbitrarily $3 < Mw < 10$ kDa), which requires non-standard approaches in chromatography and fragmentation, glycopeptide analysis may be considered an explicit form of middle-down proteomics[14–16].

The co-occurrence of PTMs can also be investigated by MS at the intact protein level[17,18]. With the advent of high mass accuracy and resolving power of MS technologies, it has now become possible to detect and baseline resolve different proteoforms directly from intact proteins under pseudo-physiological native conditions, using a modified Orbitrap mass analyzer with an extended mass range[19–25]. This approach has enabled the PTM analysis of the intact ovalbumin to an unprecedented depth, where 59 different proteoforms were detected, identified and quantified from a single-shot experiment[26]. It is of great advantage that native MS provides a direct view on the relative abundance and overall PTM composition of different co-appearing proteoforms that are distinguishable in mass[27,28]. Yet, the detailed information of the protein PTMs, such as site specificity, structure and stoichiometry on each modification site, cannot be directly extracted from the native MS data. The situation becomes more complicated when the protein of interest contains multiple PTM sites and types, especially when heterogeneous glycosylation is involved as well.

Here we present an integrative workflow to obtain a comprehensive picture of PTMs on proteins by integrating native MS and middle-down proteomic strategies (Fig. 1). As a proof of concept, we selected two structurally highly heterogeneous proteins, recombinant therapeutic human erythropoietin (rhEPO) and properdin, which were directly purified from human blood serum. Both proteins contain high and diverse amount of PTMs, demonstrating the applicability of our integrated approach in some of the most challenging scenarios. For both the proteins, we detected, identified and quantified hundreds of proteoforms with distinguishable masses at the intact protein level and also assessed detailed information of each PTM site by middle-down proteomic approaches. Furthermore, we performed in silico construction to simulate a pseudo native MS spectrum combining the site-specific PTM data obtained by the middle-down strategy. The resulting constructed intact-protein representation allowed us to directly compare the result from middle-down experiments with the native MS data. This led to the unambiguous discovery of three previously unreported C-glycosylation sites on properdin.

Furthermore, our study reports for the first time an extensive heterogeneity in occupancy of the C-mannosylation sites in the six thrombospondin repeat (TSR) domains on this protein.

We also characterized the structural heterogeneity of rhEPO therapeutics obtained from different manufacturers and sources. These rhEPO therapeutics often contain identical protein backbone sequence, albeit decorated with differential PTMs due to different conditions in their production. Technically, it is very important, yet challenging, to quantify the similarity of these therapeutics in a robust and reproducible way[29–31]. We here introduce a biosimilarity score to describe the level of structural similarity of three rhEPO therapeutics, demonstrating its usage in direct assessment of the similarity of therapeutic glycoproteins. This strategy is well applicable to other protein therapeutics, such as antibodies, thus providing a wide application range of the presented concept.

## Results

**Proteoform profiling of intact rhEPO by native MS analysis.** The molecular heterogeneity of rhEPO is mainly caused by O-glycosylation at S126 and N-glycosylation at three different sites; N24, N38 and N83 (Fig. 2a). When recording the full proteoform profile of intact rhEPO by high-resolution native MS, over 230 peaks were baseline resolved, and the overall PTM composition was assigned with satisfying mass accuracy (Fig. 2b and Supplementary Table 1). The rhEPO backbone sequence has a theoretical mass of 18,235.99 Da, and the protein masses (that is, the total molecular weight of the backbone and PTMs) measured by native MS possessing a range of 26,000 to 33,000 Da (Supplementary Table 1). This large mass difference demonstrates that about 40% of the mass is originated from glycosylation. The most abundant peak (measured mass: 29,888.12 Da, PTM composition: $Hex_{22}HexNAc_{19}Fuc_3Sia_{13}$) in the native MS spectrum likely comprises multiple proteoforms exhibiting mass alike, owing to either the combinatorial arrangement of multiple PTMs on different sites, adding up to exactly the same total mass, and/or different PTM compositions that are very close in mass. The high S/N ratio ($\sim 11,000$) allows us also to distinguish and identify lower abundant PTMs, such as the minor modifications of sialic acid by O-acetylation and $–CH_3$ to $–CH_2OH$ replacement, as illustrated in Fig. 2b.

Next, we also measured the sialidase-treated rhEPO by native MS approach. The enzymatic removal of sialic acid residues resulted in a pronounced simplification of the structural variability of rhEPO proteoforms (Fig. 2c), implying that the heterogeneity of rhEPO is mainly due to the extensive decoration with variable amounts of sialic acids. After the enzymatic treatment, the most abundant $m/z$ peak shifted to 26,102.55 Da (measured mass), which exactly corresponds to a loss of 13 sialic acids. This provides further evidence to our initial composition assignment ($Hex_{22}HexNAc_{19}Fuc_3Sia_{13}$) in the non-treated rhEPO sample.

**Site-specific PTM analysis of rhEPO by middle-down proteomics.** To characterize each modification site in detail, we digested rhEPO with the aim to separate all PTM sites into individual (glyco) peptides with suitable length by middle-down proteomics. Following careful optimization, we selected trypsin to assess the N83 and S126 glycosylation sites, and Glu-C to examine the N24 and N38 glycosylation sites. In the liquid chromatography/mass spectrometry (LC/MS) experiment, the peptide mixture was subjected to higher-energy collisional dissociation (HCD) LC/MS2 analysis. To obtain extensive fragment ions of both glycan and peptide moieties of glycopeptides, additional collision-induced dissociation (CID) and electron-transfer and higher-energy collision dissociation

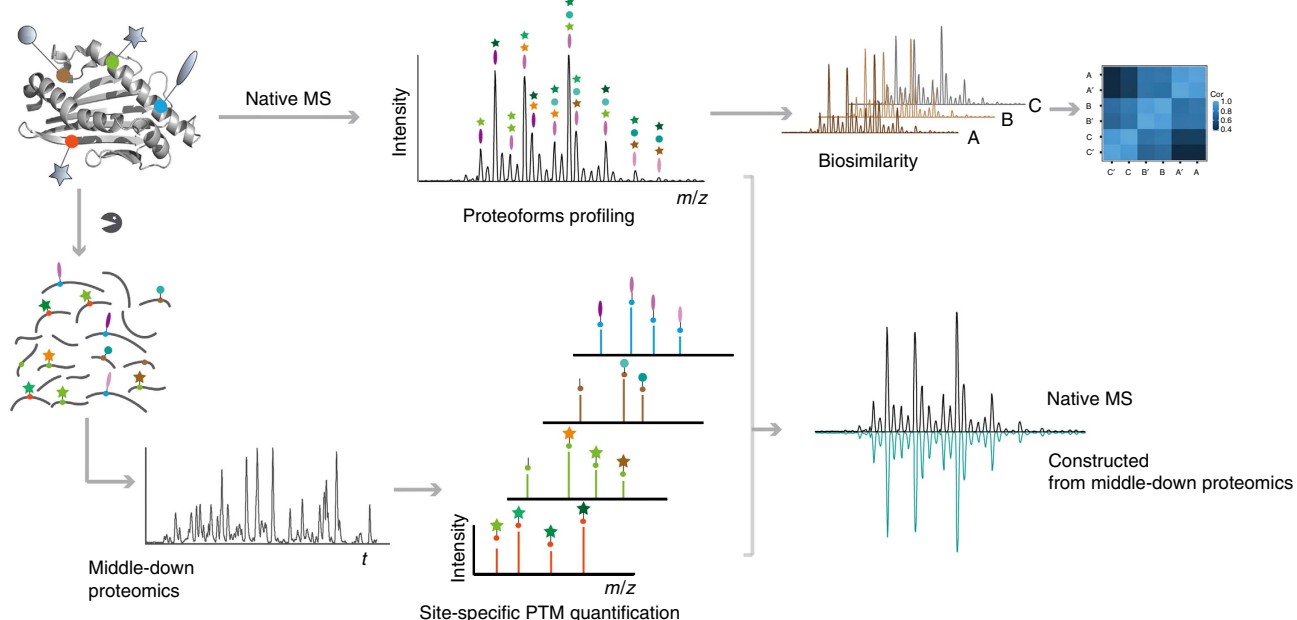

**Figure 1 | Schematic workflow of the hybrid MS approaches for the structural analysis of protein heterogeneity.** High-resolution native MS is used to qualitatively and quantitatively profile all co-appearing proteoforms and middle-down proteomic experiments are performed to characterize each modification site. The data from the site-specific PTM characterization obtained by middle-down proteomics is used to construct a native MS spectrum. A standard Pearson correlation test is conducted between the constructed and experimental native MS spectra, or between different native MS spectra to quantitatively assess the structural similarities.

(EThcD) fragmentation were applied when monosaccharides and/or disaccharides were detected in the HCD spectrum ('Methods' section). CID spectra were used to sequence the glycan branches while EThcD spectra were used to improve the sequence coverage of the peptide moiety. We also assessed the relative abundances of the differentially modified isoforms in a site-specific manner, based on the extracted ion chromatograms (XICs) of the identified peptides ('Methods' section).

As a result, we identified and relatively quantified (i) 10 glycoforms on N24, (ii) nine glycoforms on N38, (iii) eight glycoforms on N83 and (iv) two glycoforms on S126 (Fig. 3). For the O-glycosylation site on S126, we also detected the non-modified peptide. Our data revealed that each N-glycosylation site of rhEPO was modified uniquely in terms of both the numbers and relative abundances of differentially modified glycoforms. Similar to the native MS spectrum, we also observed additional modifications of NeuAc in our proteomic experiments, evidenced by multiple additional peaks ($+16$ Da, $+42$ Da or the combination of the two) next to the main glycoforms. It is known that NeuAc can be substituted with NeuGc by $-CH_3$ to $-CH_2OH$ replacement ($+16$ Da; ref. 32). In addition, both NeuAc and NeuGc can be modified with O-acetylation ($+42$ Da; ref. 33). Interestingly, we noticed that the NeuAc modification patterns are distinctive at different glycosylation sites. For instance, we detected peaks generated from both NeuGc and O-acetylation at N38 and N83; however only from NeuGc at N24. We do not have a clear biological explanation, but these observed differences could result from dissimilarities of synthetic pathways involved and/or distinct solvent accessibilities of each glycosylation site.

**Integrating native and middle-down MS for PTM profiling.** As described, we applied two complementary approaches to characterize the complex PTM profile of rhEPO. Next, we set out to further develop an integrative strategy to combine the

information from both approaches. Briefly, given the masses and relative abundances of all PTM isoforms at each site, we constructed a native protein spectrum *in silico* based on a probability model, assuming all modifications are independent events (see 'Methods' section for detailed description). This spectrum is a representation of all possible proteoforms that may exist in the sample, therefore should comprise all proteoforms detected in the native MS experiment. Positional isomers, which exhibit the same total mass, cannot be distinguished in this scenario. In this regard, we were able to compare the data from the two independent experiments and further assess the integrity of the middle-down PTM assignments.

Using the middle-down data from all PTM sites of rhEPO, we successfully constructed an intact protein spectrum that largely resembled the experimental native MS spectrum (Fig. 4a). The Pearson correlation coefficient of the two spectra was 0.86, indicating a high similarity of the two independent approaches. On the basis of the middle-down data, we further simulated a spectrum wherein all the sialic acids were removed, and compared it with the native MS data acquired from rhEPO sample treated by sialidase. In this spectrum pair (Fig. 4b), the Pearson correlation coefficient increased from 0.86 to 0.94, and nearly all proteoforms in the native MS spectrum can be annotated to the corresponding peaks in the constructed spectrum. It is known that in bottom-up and middle-down analysis, glycopeptides may easily lose their labile sialic acid moiety during sample preparation and ionization[34,35]. At the intact protein level, few studies are available regarding to this question. Rosati *et al.*[21] analysed four IgG4 hingeless mutants that exhibiting complex glycosylation including di-sialylation and tri-sialylation, and they observed that after removing sialylation, the charge state envelopes were not shifted compared with the sialylated samples. Therefore, under the conditions of native MS analysis, these labile groups remain attached to the protein, and also at the intact protein level the negatively charged sialic acid groups introduce less ionization bias[21]. This is likely a cause

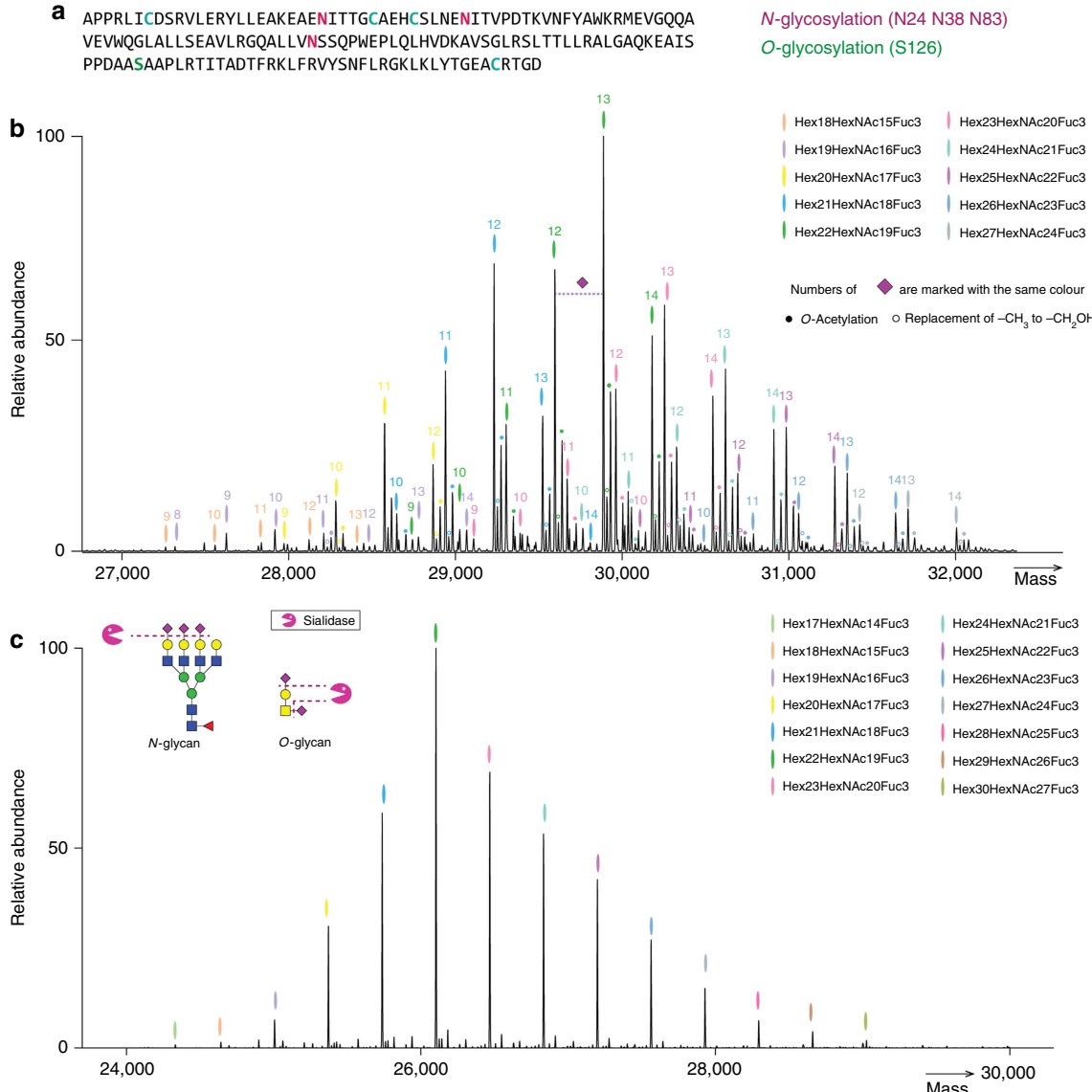

**Figure 2 | Native MS analysis of rhEPO.** (**a**) Schematic of the rhEPO backbone sequence and its reported PTM sites (Uniprot code: P01588). (**b**) The zero-charge deconvoluted native MS spectrum of rhEPO. The overall PTM compositions are assigned based on the accurate mass measurements of the intact protein. Each colour represents a glycan composition without sialic acids; the number of sialic acids attached are marked on top of each peak with the same colour. For example, the most abundant peak are marked in green and number 13, it corresponded to the glycan composition $Hex_{22}HexNAc_{19}Fuc_3Sia_{13}$. The additional $+42\,Da$ modifications represent the further $O$-acetylation of sialic acids (shown in filled circles), whereas the $+16\,Da$ modification indicates the replacement of NeuAc for NeuGc (shown in empty circles). (**c**) The zero-charge deconvoluted native MS spectrum and the overall PTM composition assignments rhEPO subjected to treatment by sialidase, which cleaves off all sialic acids present on rhEPO.

for the lower correlation observed on the unprocessed rhEPO, when compared with the sialidase-treated sample.

**Unanticipated PTM features in the plasma protein properdin.** We next studied the plasma protein properdin, also named as Factor P[36,37], representing one class of important glycoproteins. Properdin is a member of the complement protein family and plays a critical role during complement activation[38]. It has been suggested that the PTM profile of properdin derived from blood serum and neutrophil/mast cell may differ from each other, resulting in different biological outcomes[39]. The high and diverse amount of PTM sites (over 20 potentially modified sites), including $N$- and $O$- glycosylation, as well as $C$-mannosylation, make properdin an interesting but challenging target for

analysis (Fig. 5a). Specifically, it contains 6 thrombospondin repeat domains (TSR1-TSR6), known to be substrates for $C$-mannosylation, an unusual type of protein glycosylation whose exact function are yet not well understood[40]. We firstly performed native MS and middle-down proteomics to comprehensively characterize its PTM heterogeneity. In the native MS spectrum, we detected more than 30 peaks, and assigned the overall PTM composition for the significant ones (Fig. 5b). Next, we treated properdin with PNGase F, to remove the $N$-glycans. The resulted spectrum confirmed that the heterogeneity of properdin originates mainly from $C$-mannosylation, not $N$-glycosylation (Supplementary Fig. 1). Although properdin contains over 20 PTM sites, we were able to assign most of the peaks in the high resolution native MS spectrum. The extent of $N$-glycosylation, which is usually the main cause of glycoprotein heterogeneity, was

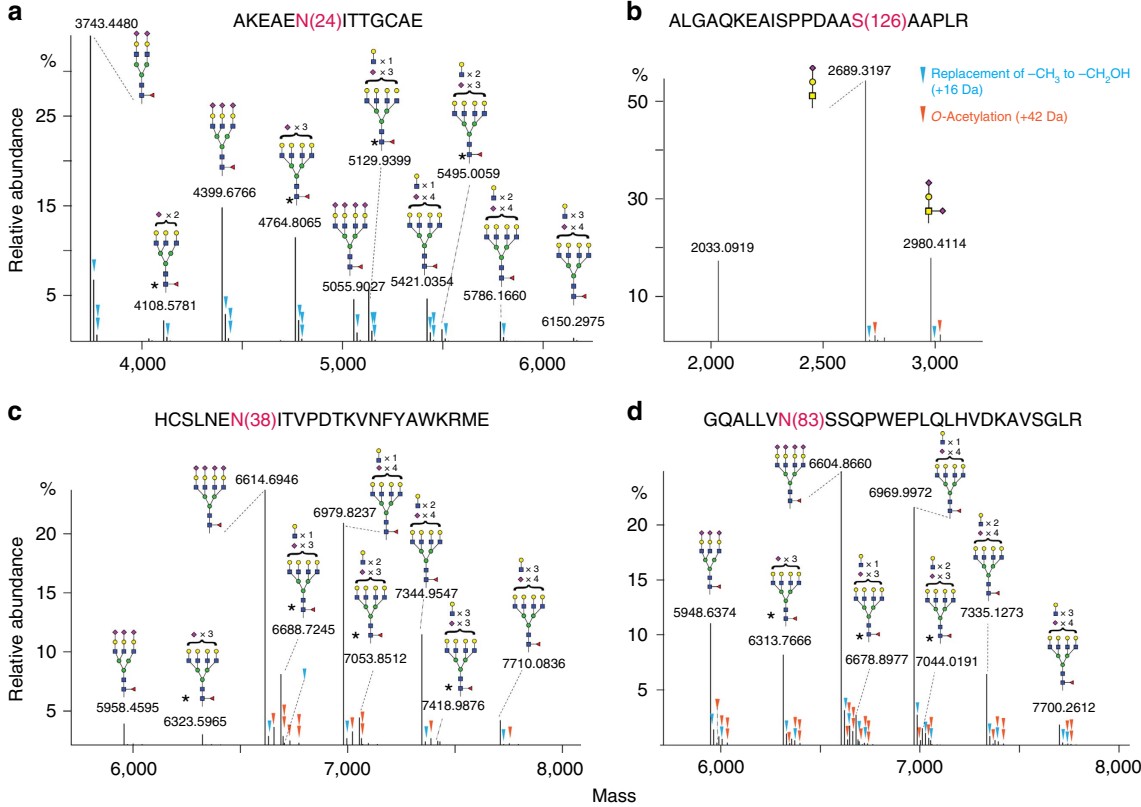

**Figure 3 | Middle-down proteomic analysis of rhEPO.** PTM profiling of site-specific glycoforms of rhEPO, covering the three *N*-glycosylation sites N24 (**a**), N38 (**c**), N83 (**d**) and the *O*-glycosylation site S126 (**b**). Of note, all the presented triantennary *N*-glycans may contain two types of structural isomers: α1–3 triantennary (shown in the figure), and α1–6 triantennary structures. * indicates the possibilities of a second co-existing glycan structure in addition to the one shown in the figure. For example, a triantennary structure with two sialic acids can also be a diantennary structure with an extra Galβ-1,4GlcNAc unit; a tetra-antennary structure with three sialic acids can also be a triantennary structure with an extra Galβ-1,4GlcNAc unit. The triangles indicate additional modification of sialic acids by either *O*-acetylation (in orange) or the replacement of −CH$_3$ by −CH$_2$OH (in blue).

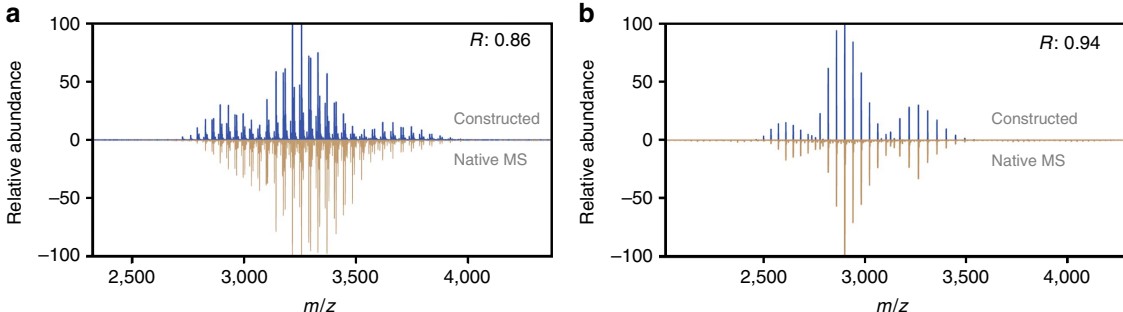

**Figure 4 | Integrative analysis of rhEPO using both high-resolution native MS and middle-down proteomics data.** (**a,b**) Comparison of the *in silico* constructed spectrum based on middle-down proteomic data (shown in blue) with the experimental native MS spectrum (shown in brown) before (**a**) or after (**b**) sialidase treatment. The Pearson correlation coefficient between the displayed spectra is shown at the top right.

found to be rather homogeneous on properdin. We demonstrated that our approach can analyse highly heterogeneous proteins and therefore its application for other plasma proteins may provide new insights on their PTM composition in unprecedented details and confidence comparing with currently used strategies. In the middle-down experiments, we identified and relatively quantified a variety of site-specific PTM isoforms originating from a staggering number of 17 *C*-mannosylation sites, four *O*-glycosylation sites and one *N*-glycosylation site (Fig. 5a). Three out of 17 identified *C*-mannosylation sites (W80, W202 and W318) have not been previously reported. The MS2 spectra of the three peptides

associated with the new sites provide a good confidence of the type of modification at these sites (Supplementary Fig. 2).

To further validate the three newly discovered *C*-mannosylation sites and the integrity of our middle-down PTM identification, we examined the middle-down data using the integrative approach described above. To this end, we constructed the native MS spectrum of properdin with or without the three newly discovered *C*-mannosylation sites. We clearly observed that, excluding the three novel *C*-mannosylation sites, several peaks were absent in the constructed spectra missed peaks in comparison with the native MS spectrum. The mass difference of 162 Da between the adjacent

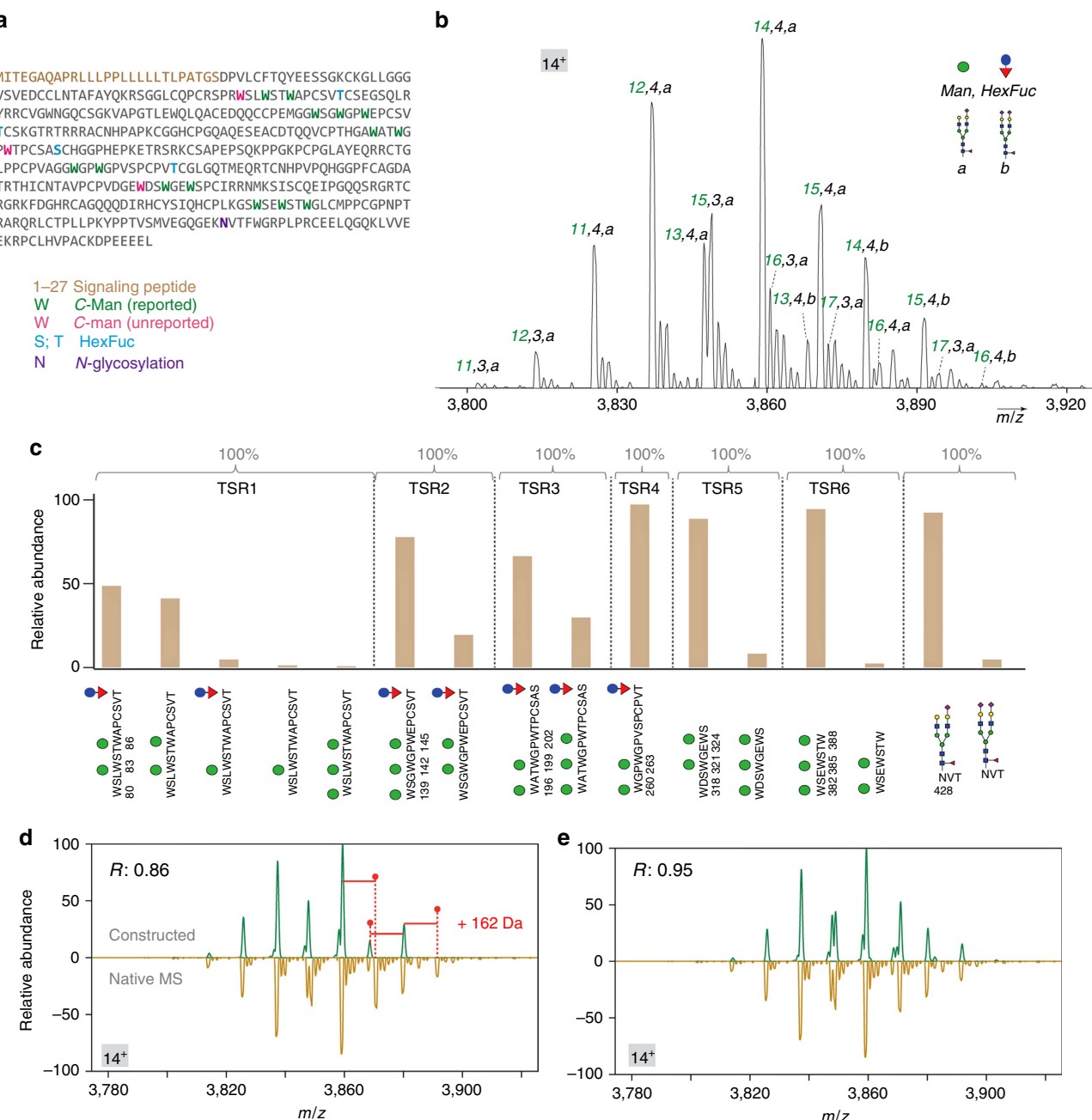

**Figure 5 | Comprehensive PTM characterization of properdin.** (**a**) The properdin sequence and its known and novel PTM sites, wherein the three newly discovered mannosylation sites at W80, W202 and W318 are annotated in red. (**b**) The native MS spectrum of properdin, zoomed in at charge state $[M + 14H]^{14+}$. The overall PTM composition is annotated. (**c**) Relative abundances of peptide proteoforms were estimated from their XICs. Each PTM-modified peptide was normalized individually so that the sum of all its proteoform areas was set at 100%. For clarity, only parts of the peptide sequence with PTMs are shown below the graph. A more detailed look at the graph in **b** shows a certain variability of C-mannosylation occupancy in different TSRs; details on this heterogeneity are described in Supplementary Note 1. (**d,e**) The comparison of the constructed spectra without (**d**) or with (**e**) the three newly discovered C-mannosylation sites with the experimental native MS spectrum of properdin. The Pearson correlation coefficient between the displayed spectra is shown at the top left.

peaks at $m/z$ range 3,860–3,900 implied the presence of three additional C-mannosylation sites at the intact protein level (Fig. 5d). After incorporating the three C-mannosylation sites in the constructed spectrum, as shown in Fig. 5e, the spectra pair looked more similar and also provided a higher Pearson correlation coefficient (0.95).

In contrast to previously reported data[41], where all the C-mannosylation sites were suggested to be fully occupied, our data reveal the existence of high heterogeneity in site occupancy

in five out of the six TSR regions. On TSR1, TSR2, TSR3, TSR5 and TSR6, the second tryptophan in the recognition motif W-X-X-W-X-X-W was found to be fully C-mannosylated, while the first and the third tryptophan were modified only partially with different level of occupancy. A more detailed description on C-mannosylation heterogeneity in the TSR structures of properdin is described in Fig. 5c and Supplementary Note 1. To our knowledge, this is by far the most thorough description of the structural heterogeneity occurring in plasma-derived properdin.

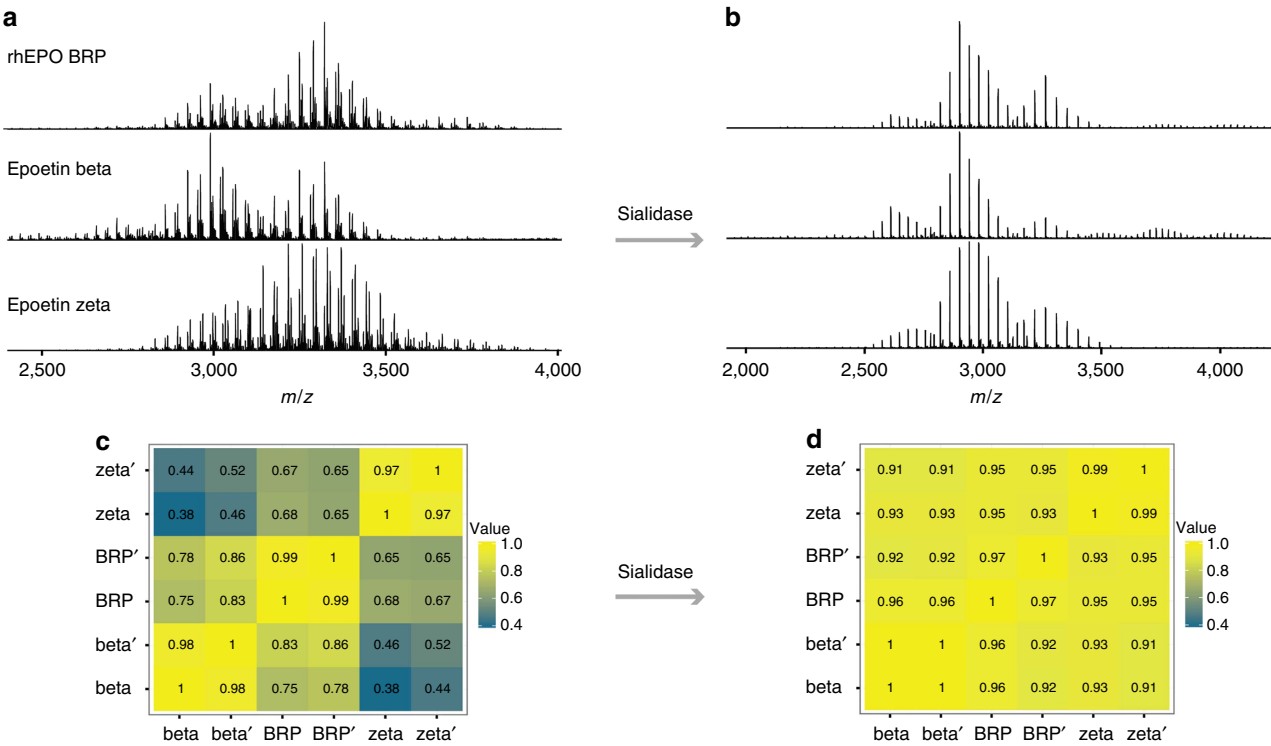

**Figure 6 | Quantitatively assessing the similarity of different rhEPO therapeutics using high-resolution native MS.** (**a,b**) Native MS spectra of three rhEPO products (rhEPO BRP, epoetin beta and epotein zeta) without (**a**) or with (**b**) sialidase treatment. (**c**) A heat map displaying spectra similarity between the three different rhEPO products. The numbers displayed in the heat map represent the Pearson correlation coefficients. Experimental duplicates are marked as beta and beta', BRP and BRP', zeta and zeta'. After removing sialic acids contents by sialidase, the Pearson correlation coefficients between EPO products increased, as shown in the heat map of desialylated EPO samples (**d**).

**Assessing biosimilarity in rhEPO samples**. The approval of a biosimilar product requires a thorough characterization, which requires the compatibility with the reference product in quality, safety and efficacy. Protein-based biosimilars often have the same backbone sequence, but may embed heterogeneity in their PTM profiles, which challenges current similarity analysis. This heterogeneity results from differences in production cell lines, culture conditions and production methods. To address the need for distinguishing biosimilar products analytically, we next demonstrate how high-resolution native mass spectrometry and our Pearson correlation-based algorithm can be used to define spectra similarity between different rhEPO products, taking three available rhEPO samples as model systems; rhEPO biological reference (BRP), epoetin beta and epoetin zeta (a biosimilar of epoetin alpha). Their respective native MS spectra are shown in Fig. 6a.

First, to assess the reproducibility of our native MS approach, we measured the correlations of technical replicates of each rhEPO sample. Our results showed all the spectral correlations were above 0.97, indicating a robust analytical reproducibility of our approach (Fig. 6c, measurement of duplicates are marked as beta and beta', BRP and BRP', zeta and zeta'). Next, we compared the samples from the different sources and observed clear differences in terms of the relative abundance of each *m/z* peak (Fig. 6a). The correlation between epoetin beta and epoetin zeta is relatively low (∼0.45), implying the distinct PTM profiles of these two rhEPO products. Notably, this observation is in agreement with a previous study that reported distinctive glycosylation patterns on epoetin alpha (a biosimilar of epoetin zeta) and beta[42]. The rhEPO BRP and epoetin beta were found to correlate better (∼0.75) than epoetin beta and zeta (∼0.45), so do rhEPO BRP and epoetin zeta (∼0.67). The observed order in

these correlations were somewhat expected since rhEPO BRP contains a mixture of epoetin alpha and beta in equal amounts (*w/w*; ref. 43).

Next, we treated the three rhEPO samples with the enzyme sialidase. This resulted in an extensive reduction of the heterogeneity in the native MS spectra (Fig. 6b). Concomitantly, the correlation increased to 0.92 between desialylated epoetin beta and zeta, 0.94 between desialylated epoetin beta and rhEPO BRP, and 0.95 between desialylated epoetin zeta and rhEPO BRP (Fig. 6d). These results demonstrate that the PTM heterogeneity of rhEPO products is largely originating from the variability in the extent and occupancy of sialylation on the various glycan trees occurring in rhEPO.

## Discussion

We applied middle-down proteomics, native MS and an integrative strategy to comprehensively assess the PTM features of rhEPO and the plasma derived properdin. Using high-resolution native MS, we obtained a qualitative and quantitative view of all co-appearing proteoforms. By using a complementary middle-down proteomic approach, we in detail revealed PTM localizations, relative abundances and glycan structures in a site-specific manner. Beneficially, we were able to assess the reliability and integrity of our PTM assignments by a direct comparison of the data from both approaches, leading us to the discovery of three new *C*-mannosylation sites as well as the undescribed structural heterogeneity in site occupancy in properdin. This integrated comparison bridges the two MS platforms, adding a new layer towards the completeness and confidence of PTM assignments.

In current analytical workflows, the characterization of structural similarity between rhEPO products poses a great

challenge, largely owing to the introduction of bias during sample preparation, analytical processes and the lack of resolving power. Here, we show that high-resolution native MS, which requires minimal sample preparation and analysis time, nicely addresses this challenge by providing a quantitative fingerprint of the intact protein. Particularly, native MS is extremely powerful at quantifying the distinct PTM patterns of glycoproteins with heavy sialylation. These monosaccharides are substantially important for the biological activities and the *in vivo* circulatory half-life of many biopharmaceutical products, such as rhEPO[44,45], yet are very challenging for quantification because they introduce ionization bias to the sialylated glycopeptide and quite fragile during sample preparation.

The hereby presented high-resolution native MS analysis and biosimilarity scoring could also be applied to other glycoprotein biologics, such as therapeutic antibodies, which exhibit typically less heterogeneous PTM profiles compared with rhEPO. Of interest, in biotechnological applications, glycosylation of glyco-proteins becomes more and more directed by perturbing specific glycosyltransferases in the host cell[46–48]. Such approaches lead to glycoproteins decorated by a plethora of varied PTMs. The effect of deletion of glycosyltransferases on the produced product could be efficiently monitored and characterized by the hereby-presented strategies.

The integrative approach offers a direct measurement of the molecular heterogeneity and can be in theory applied to any protein with PTM decoration. Plasma glycoproteins form some of the most relevant protein biomarker targets. Next to changes in abundance levels of such proteins, changes in PTM/glycosylation profiles may provide even stronger biomarker signatures[49–52]. The hereby-presented combined approach united high-resolution native mass spectrometry and middle-down proteomics for the structural analysis of glycoproteins and may, therefore, also find applicability in a variety of biotherapeutical as well as biotechnological applications.

## Methods
**Materials.** The recombinant rhEPO BRP (presently batch 4.0) was acquired from EDQM (European Directorate for the Quality of Medicines). It is a 50:50 blending of epoetin alpha and epoetin beta[53], supplied at 13,000 IU per vial. Epoetin beta was acquired from Roche and supplied at 30,000 IU per syringe. It is a recombinant form of erythropoietin marketed under the trade name NeoRecormon, produced by recombinant DNA technology using Chinese hamster ovary cells as expression system. Epoetin zeta (Retacrit, Hospira) is a biosimilar to the reference medicinal product Eprex (Epoetin alpha), produced in Chinese hamster ovary cells. The Uniprot code for rhEPO is P01588. Human blood plasma properdin (Uniprot code: P27918) was obtained from Complement Technology, Inc. (TX, USA). Urea, dithiothreitol (DTT) and iodoacetamide were purchased from Sigma-Aldrich (Steinheim, Germany). Formic acid (FA) was purchased from Merck (Darmstadt, Germany). Acetonitrile (ACN) was purchased from Biosolve (Valkenswaard, The Netherlands). Sequencing grade trypsin was obtained from Promega (Madison, WI, USA). Glu-C, Asp-N, PNGase F and Sialidase were obtained from Roche (IN, USA).

**Sample preparation for native MS.** Unprocessed protein solutions (containing 25–30 µg of the protein) were buffer exchanged into 150 mM aqueous ammonium acetate (pH 7.5) by ultrafiltration (vivaspin500, Sartorius Stedim Biotech, Germany) using a 5 kDa cut-off filter. Protein concentration was measured by ultraviolet absorbance at 280 nm and adjusted to 2–3 µM before native MS analysis. Sialidase was used to remove sialic acid residues from rhEPO. PNGase F was used to cleave the *N*-glycans of rhEPO and properdin, four units of PNGase F with 25 µg of purified protein sample (concentrated ~0.5 mg ml$^{-1}$) were mixed and incubated at 37 °C overnight. All the samples were buffer exchanged to 150 mM ammonium acetate (pH 7.5) before native MS measurements. Typically 2 µl of the concentrated sample (2 µM) were directly infused into the mass spectrometer, and the leftover from the native MS measurement can be recovered and stored if needed.

**Native MS analysis.** Samples were analysed on a modified Exactive Plus Orbitrap instrument with extended mass range (Thermo Fisher Scientific, Bremen, Germany) using a standard *m/z* range of 500–10,000 Th. The voltage offsets on transport multi-poles and ion lenses were manually tuned to achieve optimal

transmission of protein ions at elevated *m/z*. Nitrogen was used in the HCD cell at a gas pressure of $6–8 \times 10^{-10}$ bar. MS parameters used: spray voltage 1.2–1.3 V, source fragmentation 30 V, source temperature 250 °C, collision energy 30 V and resolution (at *m/z* 200) 17,500. The instrument was mass calibrated using CsI clusters.

**Native MS data analysis.** The raw spectra were deconvoluted to zero-charge spectra using the Bayesian Protein Reconstruct tool from BioAnalyst ver. 1.1 (ABSciex). The mass range and *m/z* range were adjusted accordingly. Average masses were used for PTM calculation, including hexose/mannose/galactose (Hex/Man/Gal, 162.1424 Da), *N*-acetylhexosamine/*N*-acetylglucosamine (HexNAc/GlcNAc, 203.1950 Da), deoxyhexose (dHex, 146.1430 Da), *N*-acetylneuraminic acid (Neu5Ac, 291.2579 Da), *N*-glycolylneuraminic acid (Neu5Gc, 307.2573 Da), phosphorylation (Pho, 79.9799 Da), acetylation (Acetyl, 42.0373 Da) and –CH$_3$ to –CH$_2$OH replacement (Hydroxyl, 15.9994 Da).

**Proteolytic digestion for middle-down proteomics.** Proteins (rhEPO, and properdin) were reconstituted or buffer exchanged to PBS (phosphate-buffered saline, pH 7.4) and protein concentration was adjusted to 1 mg ml$^{-1}$. Subsequently, the samples were reduced with 4 mM DTT at 56 °C for 30 min and alkylated with 8 mM iodoacetamide at room temperature for 30 min in the dark, whereafter 4 mM DTT was added again. rhEPO was digested with either trypsin at an enzyme-to-protein-ratio of 1:100 (*w/w*) or Glu-C at an enzyme-to-protein ratio of 1:75 (*w/w*) overnight at 37 °C. Properdin was digested for 4 h with either Glu-C or Asp-N at an enzyme to protein-ratio of 1:75 (*w/w*) and further treated with trypsin (1:100; *w/w*) overnight at 37 °C. For glycopeptide analysis, we typically digested 3 µg of protein for digestion, and injected 1 pmol of the resulted peptide mixture on the columns.

**LC/MS analysis.** After proteolytic digestion, all peptides (including glycopeptides) were separated and analysed using an ultra HPLC Proxeon EASY-nLC 1000 system (Thermo Fisher Scientific, Odense, Denmark) coupled online to an Orbitrap Fusion mass spectrometer (Thermo Fisher Scientific, Bremen, Germany). Reversed-phase separation was accomplished using a 100 µm inner diameter 2 cm trap column (in-housed packed with ReproSil-Pur C18-AQ, 3 µm; Dr Maisch GmbH, Ammerbuch-Entringen, Germany) coupled to a 50 µm inner diameter 50 cm analytical column (in-house packed with Poroshell 120 EC-C18, 2.7 µm; Agilent Technologies, Amstelveen, The Netherlands). Mobile-phase solvent A consisted of 0.1% FA in water, and mobile-phase solvent B consisted of 0.1% FA in ACN. The flow rate was set to 100 nl min$^{-1}$. A 50 min gradient (7–30% solvent B within 31 min, 30–100% solvent B within 3 min, 100% solvent B for 5 min, 100–7% solvent B within 1 min and 7% solvent B for 10 min) was used. For rhEPO, MS2 HCD was performed with normalized collision energy of 35%. Product ion trigger was enabled to additionally trigger CID and EThcD MS2 acquisitions of the same precursor ion if glycan masses (HexNAc: 204.0867; HexNAc fragment: 138.0545; HexNAcHex: 366.1396) were observed. CID was performed with a normalized collision energy of 25% and EThcD was performed using calibrated charge dependent ETD parameters with an EThcD supplementary activation collision energy of 20%. This allowed additional fragmentation information of both the glycan and the peptide moieties of glycopeptides. For properdin, MS2 EThcD were acquired. HCD was performed with normalized collision energy of 15% and 35%, respectively. A supplementary activation energy of 20% was used for EThcD. All the spectral data were acquired in the Orbitrap Fusion mass analyzer. For the MS scans, the scan range was set from 375 to 2,500 *m/z* at a resolution of 60,000 and the AGC target was set to $1 \times 10^6$. For the MS2 scans, the resolution was set to 15,000; the AGC target was set to $1 \times 10^5$; the precursor isolation width was 1.6 Da and the maximum injection time was set to 300 ms. The CID normalized collision energy was 35% and the ETD AGC target was set to $1 \times 10^5$.

**LC/MS data analysis.** The raw data were analysed by Byonic software (Protein Metrics Inc.) using the following parameters: precursor ion mass tolerance, 10 p.p.m.; product ion mass tolerance, 20 p.p.m.; fixed modification, Cys carbamidomethyl; variable modification: Met oxidation, STY phosphorylation, and both *N*- and *O*- glycosylation from mammalian glycan databases; allowed number of mis-cleavages: 3. For searching of the peptides we used either rhEPO or prop-erdin protein sequences. For site-specific relative quantification, the XICs were calculated using Skyline[54]. Each peptide that contains PTM sites was normalized individually so that the sum of all its proteoform areas was set at 100%. The glycan structure of each glyco-isoform was manually annotated on the basis of mass, MS2 fragmentation and confirmed as the most likely structure in literature[8,32,33,55,56]. Using MS2 fragmentation pattern, it is not possible to identify the linkage type of glycan units, such as α2–3 and α2–6 linked sialic acid, or Galβ1-4GlcNAc and Galβ1-3GlcNAc. Therefore, we did not include any linkage information in our reported glycan structures. All MS2 spectra provided fragments, which unambiguously confirmed identity of all modified peptides, type and composition of PTM.

**Middle-down data construction of pseudo-native MS spectrum.** We perform *in silico* data construction to simulate a native MS spectrum based on the masses and relative abundances of all site-specific PTM information derived from middle-down proteomics (The algorithm is made publicly available and supplemented at https://github.com/Yang0014/glycoNativeMS). The data construction is achieved on the basis of the following three elements: (1) the mass of the protein backbone retrieved from the protein sequence, (2) the masses of the PTMs on each modification sites as extracted from the middle-down data and (3) the relative abundances of site-specific PTMs extracted from LC chromatogram.

In detail, we first calculate the masses of all proteoforms based on all PTM identified from middle-down measurements. The mass of a proteoform $M$ can be calculated as:

$$M = M_{pp} + \sum_{i=1}^{n} m_{ij}$$

where $M_{pp}$ is the mass of the polypeptide portion of a given protein, which is calculated using the backbone amino acid sequence corrected for disulfide bridges; $n$ is the number of PTM sites. $m_{ij}$ is the mass of the $j$-th modification at site $i$.

Second, we calculate the relative abundance $P$ of a proteoform determined by its unique PTM combination, which is calculated by:

$$P = \prod_{i=1}^{n} P_{ij}$$

where $P_{ij}$ is the normalized relative abundance of the $j$-th modification at site $i$; $n$ is the number of PTM sites.

$$P_{ij} = \frac{A_{ij}}{\sum_{j=1}^{k} A_{ij}}$$

where $k$ is the number of possible PTM isoforms at site $i$ (including unoccupied site); $A_{ij}$ is the abundance of $j$-th modification at site $i$, which is calculated based on the XIC.

To avoid artefacts possibly induced by spectra deconvolution, the constructed data are further processed to include a charge state envelope. In this regard, it can be directly compared with the unprocessed native MS spectra. The $m/z$ spectrum is generated by converting protein masses ($M$) to $m/z$ values ($Q$):

$$Q = \sum_{k=m}^{n} \frac{M + Zk}{Zk} \times Ak$$

The native mass spectrum of a given protein often has a (limited) charge envelope ranging from $[M + mH]^{m+}$ to $[M + nH]^{n+}$, therefore we also simulate the constructed data with the same charge envelope. $Ak$ is the relative abundance of $k$-th charge state $Zk$, $m \leq k \leq n$.

**Assessing the similarities of native MS spectra.** Native MS spectra (acquired in profile mode) are first pre-processed by binning data points into defined $m/z$ ranges. Subsequently, the standard Pearson correlation is performed to score the similarity of two spectra.

An optimal bin width should be as small as possible to precisely reflect the details of two MS spectra, and reflect the resolution of the experimental native MS spectra. We first select top 20 pairs of peaks in the native spectra of rhEPO BRP and epoetin zeta, and then calculate the optimal bin width $W_{optimal}$ at three different charge states ($[M + 8H]^{8+}$, $[M + 9H]^{9+}$, $[M + 10H]^{10+}$ for rhEPO), as illustrated in Supplementary Fig. 3a. The median of the 60 optimal bin width (for each peak pair, there are three optimal bin widths, in total there are 60 numbers) is used as the bin width for the correlation test.

To further verify the selection, we plot the correlation values over a series of bin widths (1 to 20 $m/z$ with a step size 0.1, shown in blue) with a local polynomial regression fitting line shown in green. The selected bin width is close to the inflection point, as illustrated in Supplementary Fig. 3b.

**Data availability.** The data that support the findings of this study are available on request from the corresponding author (A.J.R.H.).

**Code availability.** The algorithm is written in R (version 3.3.1), and it is made publicly available and supplemented in Github. https://github.com/Yang0014/glycoNativeMS.

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

## Acknowledgements

We acknowledge the support from the Netherlands Organization for Scientific Research (NWO) funding the large-scale proteomics facility *Proteins@Work* (project 184.032.201) embedded in the Netherlands Proteomics Centre. Y.Y. and A.J.R.H. are supported by the EU funded ITN project ManiFold, grant 317371. This project has received additional funding from the European Union's Horizon 2020 research and innovation programme for MSMed, under grant agreement No. 686547. V.F. and A.J.R.H. acknowledge further support by the NWO TOP-Punt Grant 718.015.003.

## Author contributions

Y.Y., F.L., V.F. and A.J.R.H. designed the research; Y.Y., F.L. and V.F. performed the experiments; Y.Y. developed the R package for data integration; L.A.H. and H.S. provided the rhEPO samples and commented to the draft; Y.Y., F.L., V.F. and A.J.R.H. wrote the paper.

## Additional information

**Competing financial interests**: The authors declare no competing financial interests.

**How to cite this article**: Yang, Y. *et al.* Hybrid mass spectrometry approaches in glycoprotein analysis and their usage in scoring biosimilarity. *Nat. Commun.* **7**, 13397 doi: 10.1038/ncomms13397 (2016).

**Publisher's note**: 

