## [Peer Review File · Nature Communications]

REVIEWERS' COMMENTS:

Reviewer #3 (Remarks to the Author):

As I have reviewed the same manuscript for its previous submission to [redacted] I have checked whether my initial comments have been addressed. Whereas the response to the comments are satisfactory, some have not been addressed/discussed in the revised manuscript. I think that the points raised are still valid and should be included in the final manuscript.

I am copying here my previous evaluation under the headings and request that the discussion be enhanced by the points raised and the relevant references.

A. This is an excellent description of the use of state of the art Mass spectrometry to characterize in detail the post translational modifications on a protein. The focus is particularly on glycosylation modifications which, because of their extensive heterogeneity are difficult to analyse both in terms of structure and quantitative amino acid site occupation.

B. The work is original and is carried out successfully in few laboratories worldwide - in fact the combination of the challenging intact mass analysis and the detailed glycopeptide assignments are an excellent example of the use of the latest technology - and the application of the biosimilarity index by the development of an algorithm to compare the theoretical mass spectrum constructed from the glycopeptide analysis with the native heterogeneous mass spectrum is unique and can be used in practice in the pharmaceutical industry in the comparability analysis of biosimilars. this is a controversial and topical area in the biotechnology industry in which this approach holds great promise.

C. The data is quite amazing in both the resolution of the proteoforms and in the discovery, validated by parallel analysis, of new sites of the unusual tryptophan C-mannosylation. Epo in particular is a very heterogeneous recombinant glycoprotein as evidenced here - and the mass spectra able to differentiate the products of the production of the same protein sequence is not only impressive technically but offers the capacity for the production of recombinant proteins to be characterized and compared quickly and accurately in the biotechnology industry.

D&E. the very approach of comparing theoretical spectra generated from real data with the actual spectrum significantly helps solve the issue of uncertainty in data interpretation. the discovery within the spectra by this approach of new, previously undescribed

glycosylation modifications validates this approach for use in the analysis of purified glycoproteins such as those increasingly being produced as drugs in the pharmaceutical industry and the danger of uncharacterized biosimilars.

"Thank you very much for confirming the value of our work."

F. suggested improvements:

- the protein concentration is given as 2-3 micromolar - how much was actually used for both the native MS and the glycopeptide experiments.

"For the native MS measurement we directly infuse the sample into the mass spectrometer, we need $\sim 2 \mu\text{L}$ but relative high concentration (2-3 μM), so in total we infuse ~ 10 picomolar sample.

Prior to native MS measurement, we need to buffer exchange the samples into native MS compatible buffer, to get rid of salt adducts. The spin filter utilized has a dead volume of $\sim 30 \mu\text{L}$, so taking this into account, we use 25-30 μg of protein. The leftover from the native MS measurement can be recovered if necessary.

For glycopeptide analysis, we used 3 μg of protein for digestion, and then inject 1 picomol of the digested peptides for each LC/MS run." COULD NOT FIND THIS IN TEXT

- although the point is made in the legend of Figure 3 that different glycan structures can be assigned to a mass and that it is pointed out that linkages in the glycans are unable to be assigned - there may be other possibilities of structure not covered by the two examples given e.g. it is possible that there are lactosamine extensions on diantennary structures rather than multiantennary branching structures as shown in the figure? Unless further substantiated I think it is only possible to give the composition of the N-glycans on the peptide without assigning the branched structures shown? Similarly on p.5 the number of 'glycoforms' quantified are actually compositions and the glycostructures are probably many more?

"That is a reasonable concern. The MS/MS spectra cannot exclude the possibilities of lactosamine elongations on diantennary structures, but this is quite unlikely to happen on rhEPO. (Bones et. Al, Anal. Chem. 2011 June 1; 83(11): 4154-4162). Most of the annotated structures are also confirmed by literature data and they are the most likely ones, we added a statement in line 399." ADDESED

-on p.7 the deduction that the negatively charged sialic acid at the intact protein level is based on a IgG reference which has little sialylation - the highly sialylated EPO may show different effects?

"Yes indeed, EPO has much more sialylation compared with IgG. There are limited reports on this issue and this is the only story we can find. To make the comparison fairer, the IgG samples used in that particular experiment were heavily glycosylated (di-sialylation and tri-sialylation), see the ref. (Rosati et al., mAbs 5:6, 917-924; November/December 2013). The charge envelop observed after desialylation of these IgG samples was not shifted, proving that most of the charges in this cases and thus likely in our case are distributed on

the protein backbone." INCLUDE IN TEXT DISCUSSION

- in the properdin analysis was dimannose substitution on the tryptophans discounted?

"Thank you for this question. Dimannose was not considered since our CID spectra of the C-mannosylated peptides showed the same fragmentation behavior that was reported previously.(Gonzales de Peredo et al. Mol Cell Proteomics, 2011, 1:11-18). If there was dimannose we would observe a different fragmentation pattern. Moreover, most of our ETHcD MS/MS spectra of C-mannosylated peptides showed fragments between particular C-man sites which were 162 Da larger (not 324 Da which would indicate a presence of dimannose)" THE LATTER IS A GOOD ARGUMENT BUT PREVIOUS REFERENCE AND POINT SHOULD BE INCLUDED IN TEXT.

- on p. 9 the assignments of the Figure 6 in the text are incorrect. Also in the legend of Figure 6 the last sentence is repeated (with a misspelling of 'heat')

"We apologize for the mistake, it has been corrected".DONE

- Figure 5 legend, c) and d) should be d) and e)

"Our apologies, it has been corrected" DONE

- have the authors tried to enrich the glycopeptides before analysis? This is necessary in complex protein mixtures but it would be interesting to see if the ionisation (relative abundances) were different? Having said that this would presumably have become evident by the theoretical MS construction..

"Yes we tried to enrich the glycopeptide using HILIC. The relative abundances were quite comparable. But the enrichment picks up only the N-glycopeptides and we could not find back the O-glycopeptides." INTERESTING OBSERVATION SHOULD BE INCLUDED.

- Fig 1 and Fig 3 - would benefit from the peptide sequence also being shown in the site specific quantification.

"That is a good idea, we have added the sequence in." DONE

G. In the introduction the work of Kelleher and Robinson on intact MS should be cited. Perhaps a point could be made about the use of this technology on a purified glycoprotein vs the limitations for the classic proteomics approach of analysis of a complex mixture of proteins and glycoproteins.

"We added the ref based on the reviewer's advice." WHICH REFERENCE ADDED? CURRENT LIMITATIONS SHOULD BE DISCUSSED

H. Abstract etc are accurate representations.

Hybrid Mass Spectrometry Approaches in Glycoprotein Analysis and their Usage in Scoring Biosimilarity

Point-by-point to the REVIEWERS' COMMENTS:

For Clarity, **in Blue** is the new comment from the reviewer and **in Red** is our response.

Reviewer #3 (Remarks to the Author):

As I have reviewed the same manuscript for its previous submission to [redacted] I have checked whether my initial comments have been addressed. Whereas the response to the comments are satisfactory, some have not been addressed/discussed in the revised manuscript. I think that the points raised are still valid and should be included in the final manuscript.

I am copying here my previous evaluation under the headings and request that the discussion be enhanced by the points raised and the relevant references.

A. This is an excellent description of the use of state of the art Mass spectrometry to characterize in detail the post translational modifications on a protein. The focus is particularly on glycosylation modifications which, because of their extensive heterogeneity are difficult to analyse both in terms of structure and quantitative amino acid site occupation.

B. The work is original and is carried out successfully in few laboratories worldwide - in fact the combination of the challenging intact mass analysis and the detailed glycopeptide assignments are an excellent example of the use of the latest technology - and the application of the biosimilarity index by the development of an algorithm to compare the theoretical mass spectrum constructed from the glycopeptide analysis with the native heterogeneous mass spectrum is unique and can be used in practice in the pharmaceutical industry in the comparability analysis of biosimilars. this is a controversial and topical area in the biotechnology industry in which this approach holds great promise.

C. The data is quite amazing in both the resolution of the proteoforms and in the discovery, validated by parallel analysis, of new sites of the unusual tryptophan C-mannosylation. Epo in particular is a very heterogeneous recombinant glycoprotein as evidenced here - and the mass spectra able to differentiate the products of the production of the same protein sequence is not only impressive technically but offers the capacity for the production of recombinant proteins to be characterized and compared quickly and accurately in the biotechnology industry.

D&E. the very approach of comparing theoretical spectra generated from real data with the actual spectrum significantly helps solve the issue of uncertainty in data interpretation.

the discovery within the spectra by this approach of new, previously undescribed glycosylation modifications validates this approach for use in the analysis of purified glycoproteins such as those increasingly being produced as drugs in the pharmaceutical industry and the danger of uncharacterized biosimilars.

"Thank you very much for confirming the value of our work."

F. suggested improvements:

- the protein concentration is given as 2-3 micromolar - how much was actually used for both the native MS and the glycopeptide experiments.

"For the native MS measurement we directly infuse the sample into the mass spectrometer, we need ~2 μ L but relative high concentration (2-3 μ M), so in total we infuse ~ 10 picomolar sample.

Prior to native MS measurement, we need to buffer exchange the samples into native MS compatible buffer, to get rid of salt adducts. The spin filter utilized has a dead volume of ~30 μ L, so taking this into account, we use 25-30 μ g of protein. The leftover from the native MS measurement can be recovered if necessary.

For glycopeptide analysis, we used 3 μ g of protein for digestion, and then inject 1 picomol of the digested peptides for each LC/MS run." **COULD NOT FIND THIS IN TEXT**

We include the information in the Methods section, Lin 334 and. Line 364.

- although the point is made in the legend of Figure 3 that different glycan structures can be assigned to a mass and that it is pointed out that linkages in the glycans are unable to be assigned - there may be other possibilities of structure not covered by the two examples given e.g. it is possible that there are lactosamine extensions on diantennary structures rather than multiantennary branching structures as shown in the figure? Unless further substantiated I think it is only possible to give the composition of the N-glycans on the peptide without assigning the branched structures shown? Similarly on p.5 the number of 'glycoforms' quantified are actually compositions and the glycostructures are probably many more?

"That is a reasonable concern. The MS/MS spectra cannot exclude the possibilities of lactosamine elongations on diantennary structures, but this is quite unlikely to happen on rhEPO. (Bones et. Al, Anal. Chem. 2011 June 1; 83(11): 4154-4162). Most of the annotated structures are also confirmed by literature data and they are the most likely ones, we added a statement in line 399." **ADDESSED**

-on p.7 the deduction that the negatively charged sialic acid at the intact protein level is based on a IgG reference which has little sialylation - the highly sialylated EPO may show different effects?

"Yes indeed, EPO has much more sialylation compared with IgG. There are limited reports on this issue and this is the only story we can find. To make the comparison fairer, the IgG samples used in that particular experiment were heavily glycosylated (di-sialylation and tri-dialylation), see the ref. (Rosati et al., mAbs 5:6, 917-924; November/December 2013). The charge envelop observed after desialylation of these IgG samples was not shifted, proving that most of the charges in this cases and thus likely in our case are distributed on the protein backbone." **INCLUDE IN TEXT DISCUSSION**

Discussion is now included in text, from line 182.

- in the properdin analysis was dimannose substitution on the tryptophans discounted?

"Thank you for this question. Dimannose was not considered since our CID spectra of the C-mannosylated peptides showed the same fragmentation behavior that was reported previously.(Gonzales de Peredo et al. Mol Cell Proteomics, 2011, 1:11-18). If there was dimannose we would observe a different fragmentation pattern. Moreover, most of our EThcD MS/MS spectra of C-mannosylated peptides showed fragments between particular C-man sites which were 162 Da larger (not 324 Da which would indicate a presence of dimannose)" **THE LATTER IS A GOOD ARGUMENT BUT PREVIOUS REFERENCE AND POINT SHOULD BE INCLUDED IN TEXT.**

Statement included in line 408, the Methods section.

- on p. 9 the assignments of the Figure 6 in the text are incorrect. Also in the legend of Figure 6 the last sentence is repeated (with a misspelling of 'heat'

"We apologize for the mistake, it has been corrected".**DONE**

- Figure 5 legend, c) and d) should be d) and e)

.

"Our apologies, it has been corrected" **DONE**

.

- have the authors tried to enrich the glycopeptides before analysis? This is necessary in complex protein mixtures but it would be interesting to see if the ionisation (relative abundances) were different? Having said that this would presumably have become evident by the theoretical MS construction..

.

"Yes we tried to enrich the glycopeptide using HILIC. The relative abundances were quite comparable. But the enrichment picks up only the N-glycopeptides and we could not find back the O-glycopeptides." **INTERESTING OBSERVATION SHOULD BE INCLUDED.**

The observation was preliminary, and a more systematic study is already in plan, therefore we decided not including it in this manuscript.

- Fig 1 and Fig 3 - would benefit from the peptide sequence also being shown in the site specific quantification.

"That is a good idea, we have added the sequence in." **DONE**

G. In the introduction the work of Kelleher and Robinson on intact MS should be cited. Perhaps a point could be made about the use of this technology on a purified glycoprotein vs the limitations for the classic proteomics approach of analysis of a complex mixture of proteins and glycoproteins.

"We added the ref based on the reviewer's advice." **WHICH REFERENCE ADDED? CURRENT LIMITATIONS SHOULD BE DISCUSSED**

We would like to focus on intact protein PTMs characterization on high resolution platform (Orbitrap analyzer) therefore, Reference from Kelleher (ref17, 18, 22,24,25,) and Robinson (Ref 23) were added;

Our goal is to unit different MS approaches at different level, for an in-depth characterization of protein PTM complexity, rather than a high-throughput analysis. When comparing to classic peptide-centric glycoproteomic approaches, the limitation of native MS in identify protein PTM complexities is discussed in Line 74.

H. Abstract etc are accurate representations.